# Chitooligosaccharide Maintained Cell Membrane Integrity by Regulating Reactive Oxygen Species Homeostasis at Wounds of Potato Tubers during Healing

**DOI:** 10.3390/antiox11091791

**Published:** 2022-09-10

**Authors:** Pengdong Xie, Yangyang Yang, Di Gong, Lirong Yu, Ye Han, Yuanyuan Zong, Yongcai Li, Dov Prusky, Yang Bi

**Affiliations:** 1College of Food Science and Engineering, Gansu Agricultural University, Lanzhou 730070, China; 2Department of Food Science, Agricultural Research Organization, The Volcani Center, Rishon LeZion 7505101, Israel; 3Department of Postharvest Science, Agricultural Research Organization, The Volcani Center, Rishon LeZion 7505101, Israel

**Keywords:** *Solanum tuberosum* L., chitooligosaccharide, wound healing, ROS generation and scavenging, cell membrane integrity

## Abstract

Chitooligosaccharide (COS) is a degradation product of chitosan. Although COS increased fruit resistance by regulating the metabolism of reactive oxygen species (ROS), few reports are available on whether COS regulates ROS homeostasis at wounds of potato tubers during healing. In this study, COS increased gene expression and activities of NADPH oxidase and superoxide dismutase, and promoted the generation of O_2_^●−^ and H_2_O_2_. Moreover, COS increased gene expression and activities of catalase, peroxidase, and AsA–GSH cycle-related enzymes, as well as the levels of ascorbic acid and glutathione levels. In addition, COS elevated the scavenging ability of DPPH, ABTS^+^, and FRAP, and reduced cell membrane permeability and malondialdehyde content. Taken together, COS could maintain cell membrane integrity by eliminating excessive H_2_O_2_ and improving the antioxidant capacity in vitro, which contributes to the maintainance of cell membrane integrity at wounds of potato tubers during healing.

## 1. Introduction

Reactive oxygen species (ROS) play a crucial role in potato tubers during wound healing. ROS produced in the early stages of healing act as signaling molecules to activate a series of metabolisms related to wound healing, while ROS produced in the later stage act as an oxidant and participate in the oxidative cross-linking of suberin polyphenolic and lignin at wounds [1]. ROS in potato tubers during healing are mainly derived from NADPH oxidase (NOX) [2], which generate O_2_^●−^ by transferring electrons to O_2_, and then O_2_^●−^ is quickly dismutated into stable H_2_O_2_ by superoxide dismutase (SOD) [3].

Chitooligosaccharide (COS) is a low molecular weight product of chitosan degradation with a degree of polymerization between 2–20 and an average molecular weight less than 3900 Da [4]. COS could induce the resistance of fruit and have antifungal activity [5]. COS increased SOD activity, promoted the accumulation of H_2_O_2_, and induced the resistance of citrus fruit against *Colletotrichum gloeosporioides* [6]. Moreover, COS increased catalase (CAT) and peroxidase (POD) activities, and induced the resistance of peach fruit against *Monilinia fructicola* and *Penicillium expansum* [7]. COS up-regulated gene expression of *SOD* and *CAT* in kiwifruit and enhanced fruit resistance to *Botrytis cinerea* and *P. expansum* [8]. COS also up-regulated gene expression of *SOD*, *CAT*, and *APX*, eliminated excessive ROS [9], and increased ascorbate peroxidase (APX) activity and glutathione (GSH) contents in Dongzao fruit [10]. In addition, COS increased the scavenging capacity of H_2_O_2_ in citrus fruit by inducing APX and glutathione reductase (GR) activities and promoting the accumulation of ascorbic acid (AsA) and GSH [6]. Additionally, COS increased the scavenging ability of DPPH and ABTS^+^ in blackberry fruit [11] and improved DPPH antioxidant capacity in strawberry fruit by increasing total phenols and flavonoids contents [12]. Furthermore, COS maintained cell membrane integrity and reduced malondialdehyde (MDA) content in harvested kiwifruit [13].

Although it has been reported that COS induced fruit resistance by regulating ROS metabolism, a few reports are available on whether COS regulates ROS homeostasis and maintains cell membrane integrity at wounds of potato tubers during healing. Therefore, we hypothesize that COS may regulate the generation and scavenging of ROS, and enhance the antioxidant capacity at wounds of potato tubers during healing.

The objectives of this study were to (1) measure the gene expression and activities of NOX and SOD, as well as the O_2_^●−^ and H_2_O_2_ contents at wounds of potato tubers; (2) determine the gene expression and activities of CAT and POD at wounds; (3) assess the enzymes activity, contents of substrates, and product related to the AsA–GSH cycle at wounds; and (4) evaluate the antioxidant capacity in vitro and cell membrane integrity at wounds.

## 2. Materials and Methods

### 2.1. Materials

Potato tubers (*Solanum tuberosum* L. cv. Atlantic) were obtained from Ailan Potato Seed Industry Co., Ltd., Dingxi City, Gansu Province. Tubers with the uniform size and without visible disease and mechanical damage were selected and transported to the lab on the same day, and then placed at ambient conditions (22 ± 3 °C, RH 55–65%) for use.

Chitooligosaccharide (deacetylation ≥ 85%, purity ≥ 99%, molecular weight 1200 Da) was purchased from Shandong Hailongyuan Biotechnology Co., Ltd., Zhucheng, China.

### 2.2. Methods

#### 2.2.1. Artificial Wound of Tubers and COS Treatment

The creation of artificial wounds on tubers and the COS treatment were performed according to the approach of Zhu et al. [14]. After washing and disinfecting surfaces, tubers were wiped with 75% ethanol and then air dried. Subsequently, the tubers were cut in half along the equator with a peeler and then dipped in 5 g L^−1^ COS solution (the treated concentration was screened by the previous assay results) for 10 min. After air-drying, the tubers were kept in polyethylene bags (25 cm × 40 cm, 0.02 mm thickness) and placed in the dark (22 ± 3 °C, RH 55–65%) for wound healing. The tubers treated with distilled water were measured as the control.

#### 2.2.2. Determination of Cell Membrane Permeability

Cell membrane permeability was measured according to the method of Jiang et al. [1]. Nine discs (diameter 10 mm, thickness 3 mm) were taken from wounds and washed with deionized water. Then, the discs were incubated in 40 mL deionized water, and conductivity was immediately determined with a conductivity meter at 25 °C and recorded as C_0_. It was then recorded as C_1_ after incubating for 3 h. Finally, the discs were incubated in boiling water for 30 min, and conductivity was measured and recorded as C_F_ after cooling. The cell membrane permeability was calculated using the following formula:$$\mathrm{Cell}\;\mathrm{membrane}\;\mathrm{permeability}\;\left(\%\right)=\frac{C_{1}\;-\;C_{0}}{C_{F}}\;\times \;100\%$$

#### 2.2.3. Sampling

At 0, 3, 5, 7, 14, and 21 d of tubers healing, 2 mm thick healing tissues were taken from the surface of wounds with a scalpel. Samples were immediately ground into powder with a grinder (A11 B S025, IKA-Werke GmbH & Co.KG Co., Ltd., Staufen im Breisgau, Germany) and stored at −80 °C [15].

#### 2.2.4. Determination of MDA Content

MDA content was measured based on the method of Hodges et al. [16]. Frozen powder (1.0 g) was added with 3 mL of 100 g L^−1^ trichloroacetic acid (TCA), and homogenate was centrifuged at 12,000× *g* for 30 min at 4 °C. The supernatant was used for MDA analysis, and absorbance was determined at 450, 532, and 600 nm. MDA content was expressed as µmol mg^−1^ FW.

#### 2.2.5. Determination of O_2_^●−^ and H_2_O_2_ Contents

The production rate of O_2_^●−^ was measured according to the method of Zheng et al. [17]. Frozen powder (1.0 g) was added to 3 mL of 50 mM phosphate-buffered solution (0.3% Triton X-100, 20 g L^−1^ PVP, 1 mM EDTA) and the homogenate was centrifuged at 10,000× *g* for 30 min at 4 °C. The supernatant was used as a crude enzyme solution. 1.0 mL supernatant was added with 1.0 mL of PBS (50 mM, pH 7.8) and 1.0 mL hydroxylamine hydrochloride, and incubated at 25 °C for 1 h. Then, 4-aminobenzene sulfonic acid (1.0 mL, 17 mM) and α-naphthylamine (1.0 mL, 7 mM) were added. The absorbance was measured at 530 nm after incubation at 25 °C for 20 min. The production rate of O_2_^●−^ was calculated with sodium nitrite as the standard, which was expressed as μmol g^−1^ min^−1^ FW.

The H_2_O_2_ content was determined with an H_2_O_2_ assay kit (Solarbio Biotechnology Co., Ltd., Beijing, China). The H_2_O_2_ content determination was performed according to the instructions of the manufacturer at 415 nm, and expressed as mmol kg^−1^ FW.

#### 2.2.6. Total RNA Extraction and qRT-PCR Analysis

Total RNA was extracted according to the method of Kundu et al. [18]. RNA was extracted from frozen powder with Polysaccharides & Polyphenolics-rich (TianGen, DP441) and cDNA was synthesized by the PrimeScrip RT reagent Kit with the gDNA Eraser Kit (TaKaRa, RR047A). The TB Green Premix Ex Taq^TM^ II (Tli RNaseH Plus) (TaKaRa, RR820A) kit and real-time fluorescence quantitative PCR system (QuantStudioR5, THERMO FISHER) were used for quantification. Each gene was detected three times and relative expressions were calculated using the 2^−ΔΔCt^ method. All primer sequences are shown in Table 1.

#### 2.2.7. Enzyme Activities

NADPH oxidase (NOX) activity was determined according to the method of Sagi and Fluhr. [19]. Frozen powder (1.0 g) was homogenised in 5 mL Tris-MES buffer (pH 7.8) containing 250 mM sucrose, 3 mM ethylene diamine tetraacetic acid (EDTA), 0.9% polyvinyl pyrrolidone (PVP), 5 mM dithiothreitol (DTT), and 1 mM phenylmethanesulfonyl fluoride (PMSF) for 30 min. Then, the homogenates were centrifuged at 3000× *g* for 10 min at 4 °C. After removing the supernatant, the residue was centrifuged at 12,000× *g* for 40 min. The precipitate obtained was suspended in 1 mL Tris-MES buffer containing 250 mM sucrose, 5 mM KCl, 5 mM DTT, and 1 mM PMSF, and used for assaying the NOX activity. The reaction mixture contained 50 mM Tris-HCl buffer (pH 7.5), 0.5 mM XTT, 100 µM NADPH, and 30 µL enzyme extraction with the final addition of NADPH. The NOX activity is expressed as U g^−1^ FW, where U was defined as an increase of 0.01 in absorbance per minute at 290 nm.

Superoxide dismutase (SOD) activity was determined according to the method of Bai et al. [20]. Frozen powder (1.0 g) was homogenised in 3 mL 50 mM phosphate buffer (pH 7.8) for 30 min and centrifuged at 12,000× *g* for 30 min at 4 °C, and the supernatant was used for the SOD activity assay. The reaction mixture consisted of 1.5 mL 50 mM phosphate buffer (pH 7.8), 200 µL 100 mM methionine (MET), 100 µM EDTA-Na_2_, 750 µM nitro blue tetrazolium (NBT), and 200 µL supernatant, followed by 100 µL of riboflavin. After reacting under a fluorescent lamp of 4000 lx for 1 min, the absorbance was measured at 560 nm and defined through the amount of enzyme that caused a 50% inhibition of NBT and was expressed as U g^−1^ FW.

Peroxidase (POD) activity was determined according to the method of Venisse et al. [21]. Frozen powder (1.0 g) was homogenised in 3 mL 50 mM phosphate buffer (pH 7.5) (containing 1 mM polyethylene glycol (PEG), 1 mM PMSF, 8% PVPP (*w/v*) and 0.01% Triton X-100 (*v/v*)) and extracted for 30 min, then centrifuged at 12,000× *g* for 30 min at 4 °C and the supernatant collected for a POD activity assay. The reaction mixture, containing 2 mL phosphate buffer (pH 7.5), 200 μL crude enzyme, 25 mM guaiacol and 0.25 M H_2_O_2_, was maintained at 24 °C for 5 min and the absorbance was measured at 470 nm. The POD activity was expressed as U g^−1^ FW.

Catalase (CAT) activity was determined using a kit (Solarbio Biotechnology Co., Ltd.). Frozen powder (0.1 g) was added to 1 mL of the pre-cooled extract solution to homogenize in an ice bath. The homogenate was centrifuged at 8000× *g* for 10 min at 4 °C, and the supernatant was collected and added to the other reagents according to the kit manufacturer’s instructions. The CAT activity was measured at 405 nm and was expressed as U g^−1^ FW.

Ascorbate peroxidase (APX) activity was determined according to the method of Bao et al. [22]. Frozen powder (1.0 g) was homogenized with a mortar and pestle in 3 mL 100 mM phosphate buffer (pH 7.5) containing 1 mM ethylene diamine tetraacetic acid (EDTA) and centrifugated at 12,000× *g* for 30 min at 4 °C. The supernatant was used for enzyme assays. The assay mixture consisted of 2 mL of 100 mM phosphate buffer (pH 7.5), 200 mL of 3 mM ascorbic acid, 1.4 mL of 30% H_2_O_2_ (*v/v*), and 200 mL of crude enzyme extract. The absorbance was measured at 290 nm and the APX activity was expressed as U g^−1^ FW. 

Dehydroascorbate reductase (DHAR) and monodehydroascorbate reductase (MDHAR) activities were determined using a kit (Solarbio Biotechnology Co., Ltd.). Frozen powder (0.1 g) was homogenised in 1 mL of the pre-cooled extract solution to homogenize in an ice bath. The homogenate was centrifuged at 8000× *g* for 10 min at 4 °C, and the supernatant was collected and added to the other reagents according to the kit manufacturer’s instructions. The DHAR and MDHAR activities were measured at 265 nm and 340 nm, and was expressed as U g^−1^ FW.

Glutathione reductase (GR) activity was determined according to the method of Queirós et al. [23]. Frozen powder (1.0 g) was homogenised in 3 mL 0.1 M Tris-HCl (pH 7.8) extraction buffer (containing 2 mM EDTA and 2 mM DTT) and centrifugated at 12,000× *g* for 30 min at 4 °C. The supernatant was used for GR activity analysis. The reaction mixture was a total volume of 200 μL crude enzyme, 2 mM NADPH, 3 mM MgCl_2_ and 10 mM oxidised glutathione. GR activity was defined as the amount of enzyme that caused a decrease in absorbance within 120 s at 340 nm, and was expressed as U g^−1^ FW.

#### 2.2.8. Determination of Substrates and Product Contents of the AsA–GSH Cycle

Ascorbic acid (AsA) and dehydroascorbic acid (DHA) were determined according to the method of Turcsányi et al. [24]. Frozen powder (1.0 g) was added to 3 mL 100 mmol L^−1^ HCl, and the homogenate was centrifuged at 8000× *g* for 10 min at 4 °C. The AsA reaction system comprised 100 μL supernatant, 2 mL of 100 mM potassium phosphate buffer, and 0.5 mL distilled water, and absorbance was determined at 265 nm. AsA content was expressed as mg g^−1^. The total ascorbic acid reaction system comprised 100 μL supernatant, 2 mL of 100 mM potassium phosphate buffer, and 0.5 mL of 2 mM DTT, and absorbance was determined at 265 nm after reaction at room temperature for 8 min. Total ascorbic acid content minus the AsA content results in the DHA content, which was expressed as mg g^−1^.

The content of GSH was determined with a GSH assay kit (Solarbio Biotechnology Co., Ltd.). The GSH content determination was performed according to the instructions of the manufacturer at 412 nm and was expressed as mg g^−1^.

The content of GSSG was determined with a GSSG assay kit (Solarbio Biotechnology Co., Ltd.). The GSSG content determination was performed according to the instructions of the manufacturer at 412 nm and was expressed as mg g^−1^.

#### 2.2.9. Determination of the Scavenging Ability of ABTS^+^, DPPH and FRAP

The scavenging ability of DPPH, ABTS^+^, and FRAP were determined with an assay kit (Solarbio Biotechnology Co., Ltd.). The determination of the scavenging ability of DPPH, ABTS^+^, and FRAP was performed according to the instructions of the manufacturer at 515 nm, 734 nm, and 593 nm, and was expressed as μmol Trolox g^−1^ FW.

### 2.3. Statistical Analysis

The result was exhibited as mean ± standard error and three biological replicates were performed. Differences between the treatments were compared by ANOVA. Mean comparisons were performed using Duncan’s multiple range test, which examined if the differences were significant at *p* < 0.05.

## 3. Results

### 3.1. COS Activated NOX and SOD and Enhanced O_2_^●−^ and H_2_O_2_ Contents at Tuber Wounds

NOX generates O_2_^●−^ by transferring electrons to O_2_, and then O_2_^●−^ is quickly dismutated into stable H_2_O_2_ by SOD [1]. In the present study, *StNOX* expression showed single peaks at 3 d and 5 d, respectively. The COS group had higher *StNOX* expression during healing, which was 1.1-fold higher than that of the control at 7 d (Figure 1A). Likewise, NOX activity increased in all groups on the first 5 d of healing, and the COS group had higher NOX activity compared to the control during healing (Figure 1B). The *StSOD* expression in the COS group showed a single peak at 5 d, while a decreasing trend of the expression was found in the control during healing. Moreover, higher *StSOD* expression was found in the COS group during healing, which was 3.1 folds higher than that of the control at 5 d (Figure 1C). Additionally, SOD activity increased in all sample groups during healing, while the COS group maintained higher activity than that of the control except 3 d (Figure 1D).

The peak of O_2_^●−^ content was found at 5 d of healing in all groups, while the COS group maintained a higher value during healing, which was 37.6% higher than that of the control at 5 d (Figure 2A). The H_2_O_2_ content peaked at 14 d in all groups, while the COS group maintained a higher value during healing (Figure 2B).

These results suggested that COS activated NOX and SOD, and enhanced the contents of O_2_^●−^ and H_2_O_2_ at tuber wounds.

### 3.2. COS Elicited CAT and POD at Tuber Wounds

CAT and POD are important enzymes for scavenging H_2_O_2_ [25]. The expression of *StCAT* and *StPOD* in all sample groups showed a single peak at 5 d, and the COS group maintained higher levels during healing, which were 2.63 and 2.66 folds higher than the control at 7 d, respectively (Figure 3A,C). The CAT activity in all sample groups showed a single peak at 7 d, and the COS group maintained higher activity during healing (Figure 3B). An increase of POD activity was found in all sample groups during healing, while the COS group maintained higher activity, which was 55.6% higher than the control at 21 d (Figure 3D). These results suggested that COS treatment activated CAT and POD at tuber wounds.

### 3.3. COS Activated the AsA–GSH Cycle at Tuber Wounds

APX, DHAR, MDHAR, and GR are involved in the AsA–GSH cycle [26]. The expression of *StAPX*, *StDHAR,* and *StMDHAR* in all groups showed a single peak, while the COS group had higher levels during healing, which were 3.1, 2.4, and 0.8 folds higher than that of the control at 7 d of healing (Figure 4). APX activity increased in all groups during healing, while the COS group maintained higher content, which was 32% higher than that of the control at 21 d of healing (Figure 4). DHAR and MDHAR activities peaked at 7 d of healing in all groups, and the COS group kept higher values, which were 29.2% and 30.6% higher than that of the control at 21 d of healing, respectively (Figure 4). The *StGR* expression in the COS group peaked at 3 d of healing, while it decreased and then increased in the control during healing. At 7 d of healing, the *StGR* expression in the COS group was 4.3 folds higher than that of the control (Figure 4). Likewise, the peaks of GR activity in all groups were found at 7 d, while the COS group maintained a higher level, which was 38.4% higher than that of the control at 7 d of healing (Figure 4).

AsA, DHA, GSH, and GSSG are substrates and products of AsA–GSH [27]. A peak of AsA content was found at 7 d of healing in all groups, and the COS group stayed higher during healing; 40% higher than the control at 5 d of healing (Figure 5A). Likewise, the DHA content peaked at 7 d of healing in all groups, while the COS group maintained a lower value during healing; 43.9% lower than the control at 7 d of healing (Figure 5B). A trend showing an increase of GSH content was found in all groups during healing, while the COS group had a higher value, which was 25.2% higher than the control at 21 d of healing (Figure 5C). A peak of GSSG content was found at 7 d of healing in all sample groups, while the COS group had a lower value during healing, which was 24.7% lower than the control at 7 d of healing (Figure 5D).

These results suggested that COS treatment increased the activities of APX, DHAR, MDHAR, and GR, and promoted the accumulation of AsA and GSH, thereby contributing to the elimination of the excessive H_2_O_2_ at tuber wounds.

### 3.4. COS Enhanced the Antioxidant Capacity In Vitro at Tuber Wounds

The scavenging ability of DPPH, ABTS^+^, and FRAP are important indicators for evaluating antioxidant activity [28]. The scavenging ability of DPPH was higher in all sample groups, and the COS group had a higher value during healing, which was 20.4% higher compared to the control at 21 d of healing (Figure 6A). An increased scavenging ability of ABTS^+^ and FRAP were found in all groups during healing, while the COS group was higher compared to the control (Figure 6B,C). These results suggested that COS treatment enhanced antioxidant capacity in vitro at tuber wounds.

### 3.5. COS Reduced Cell Membrane Permeability and MDA Content at Tuber Wounds

Cell membrane permeability and MDA content reflect membrane integrity and the level of membrane lipid peroxidation [29]. In all groups, cell membrane permeability increased during healing, while the COS group maintained a lower value, which was 21.5% lower than the control at 21 d of healing (Figure 7A). A peak of MDA content was found in all groups, while the COS group had a lower value during healing, which was 23.8% lower than the control at 7 d of healing (Figure 7B). These results suggested that COS treatment effectively maintained cell membrane integrity and inhibited membrane lipid peroxidation at wounds of tubers.

## 4. Discussion

During the healing process of potato tubers, ROS mainly comes from NOX, POD, and PAO, while NOX is considered as the most important source [1]. NOX generates O_2_^●−^ by transferring electrons to O_2_. Then, O_2_^●−^ is quickly dismutated into stable H_2_O_2_ by SOD due to its instability [30]. In this study, COS treatment activated NOX and SOD (Figure 1), and promoted the generation of O_2_^●−^ and H_2_O_2_ (Figure 2) at wounds of potato tubers. These results are similar to the results which showed that COS activated NOX and induced H_2_O_2_ accumulation in rice leaves [31]. It has been reported that COS induced extracellular Ca^2+^ influx and increased the resistance of tobacco against the tobacco mosaic virus [32]. Moreover, COS increased the activity of calcium-dependent protein kinase (CDPK) in *Brassica napus* [33] and up-regulated the expression of *CDPK7* and *CDPK26* in *Arabidopsis* [34]. Increased concentration of intracellular Ca^2+^ activated CDPK, and the interaction of CDPK and NOX induced ROS bursts at tuber wounds [3]. Therefore, we hypothesize that COS may activate CDPK by inducing extracellular Ca^2+^ influx, primarily at wounds of tubers, and then CDPK phosphorylates NOX, which generates O_2_^●−^ and H_2_O_2_.

CAT and POD are important enzymes for scavenging H_2_O_2_ in plants. CAT can reduce H_2_O_2_ to H_2_O and O_2_ [35]. Moreover, POD also can hydrolyze H_2_O_2_ into H_2_O and O_2_ [36]. In this study, COS treatment up-regulated *StCAT* expression and increased CAT activity at wounds of tubers (Figure 3).These findings are similar to the results that found COS increased *CAT* expression and CAT activity in kiwifruit [8]. Moreover, COS treatment activated *StPOD* expression and increased POD activity at wounds of tubers. These results are similar to the results that found COS activated *POD* expression and increased POD activity in peach fruit [37]. Therefore, we hypothesized that COS treatment promotes H_2_O_2_ generation at tuber wounds during healing, which may provide substrates for CAT and POD and activate CAT and POD.

AsA–GSH is a significant ROS-scavenging system in organisms, which contributes to scavenging excess H_2_O_2_ [38]. APX is the first enzyme in AsA–GSH, which can specifically catalyze AsA to MDHA [25]. MDHAR cooperates with APX to eliminate H_2_O_2_ [39]. DHAR catalyzes DHA to AsA in the presence of GSH and NADH [26], and GR converts GSSG to GSH by the action of NADPH, which further reduces oxidative stress [40]. In this study, COS treatment activated the expression of *StAPX*, *StDHAR*, *StMDHAR,* and *StGR,* and increased activities of APX, DHAR, MDHAR, and GR at wounds of tubers (Figure 4). Moreover, the treatment also enhanced AsA and GSH contents and decreased DHA and GSSG contents (Figure 5). These results are similar to the results that found COS activated APX and GR, and increased AsA and GSH contents in citrus fruit [6]. Therefore, we hypothesized that COS treatment increases H_2_O_2_ content (Figure 2), which may provide substrates for AsA–GSH, thereby activating the AsA–GSH cycle and promoting the scavenging ability of H_2_O_2_ at wounds of tubers.

DPPH, ABTS^+^, and FRAP are important indicators for evaluating antioxidant activity in vitro. DPPH is mainly used to evaluate the antioxidant capacity of phenols, flavonoids, and terpenoids in plants [41]. ABTS^+^ and FRAP are mainly used to evaluate the antioxidant capacity of vitamins and carotenoids, respectively [42]. In this study, COS treatment increased the scavenging ability of DPPH, ABTS^+^, and FRAP at wounds of tubers, these results are similar to the results that found COS increased the scavenging ability of DPPH and ABTS^+^ in blackberry fruit [11]. An amount of phenols, flavonoids, terpenoids [43], AsA [44], polyamines, and alkaloids [45] were generated at the wounded surface of potato tubers during healing. In addition, COS has a strong free radical scavenging ability [46]. Therefore, we inferred that COS could enhance the scavenging ability of DPPH, ABTS^+^, and FRAP at tuber wounds, which may closely relate to the increase of the total phenols, flavonoids, AsA, and terpene contents at wounds, as well as to its free radical scavenging ability.

Cell membrane integrity is the key to maintaining normal metabolisms, and intact cell membranes ensure the smooth progress of primary and secondary metabolisms of tubers during healing [1]. Excessive ROS leads to membrane lipid peroxidation, which formats MDA and destroys cell membrane integrity, thereby reducing the normal metabolic levels of cells [47]. In this study, COS treatment reduced cell membrane permeability and MDA content, and maintained cell membrane integrity at tuber wounds (Figure 7). These results are similar to the results that COS reduced MDA content in peach fruit [7]. Therefore, we infer that COS activated CAT and POD (Figure 3) and AsA–GSH (Figure 4 and Figure 5), which may contribute to scavenging excessive H_2_O_2_ at tuber wounds (Figure 2). In addition, COS treatment increased the scavenging ability of DPPH, ABTS^+^, and FRAP at tuber wounds (Figure 6), which may maintain cell membrane integrity and ensure the smooth progress of normal metabolisms during healing.

As nearly almost all potato tubers are harvested mechanically, after harvesting, the tubers are handled by grading, packaging, and transportation, which can damage the tubers to varying degrees [45]. The wounds not only accelerate water evaporation, but also provide access to pathogen invasion, which causes serious decay during storage [48]. Our research indicates that COS could promote wound healing of potato tubers by depositing suberin and lignin at wounds (unpublished), and ROS homeostasis regulated by COS plays an important role in maintaining the monomers synthesis of suberin and lignin. Interestingly, potato tubers have a wound-healing ability, which contributes to the inhibition of water evaporation and helps to reduce pathogen infection by forming a periderm layer at wounds [49]. However, since natural healing of tubers takes a long time, specific techniques are needed to speed up the process [14]. In addition, COS is generally considered a chemical with safe, low-cost, and broad-spectrum effects. Therefore, COS could be developed as an environment-friendly wound healing accelerator.

## 5. Conclusions

COS activated NOX and SOD, and promoted the generation of O_2_^●−^ and H_2_O_2_ at tuber wounds. COS activated CAT, POD, and AsA–GSH to eliminate excessive H_2_O_2_ at tuber wounds. In addition, COS also increased the scavenging ability of DPPH, ABTS^+^, and FRAP at wounds, which contributed to the maintenance of cell membrane integrity and ensured the smooth progress of primary and secondary metabolisms at tuber wounds during healing. This study elucidates that COS could maintain cell membrane integrity by regulating ROS homeostasis at wounds of potato tubers. Therefore, COS can beneficially accelerate the process of wound healing and reduce postharvest loss of potato tubers during storage. The possible mode of ROS generation and scavenging in potato tubers induced by COS treatment during healing is shown in the graphical abstract.

## Figures and Tables

**Figure 1 antioxidants-11-01791-f001:**
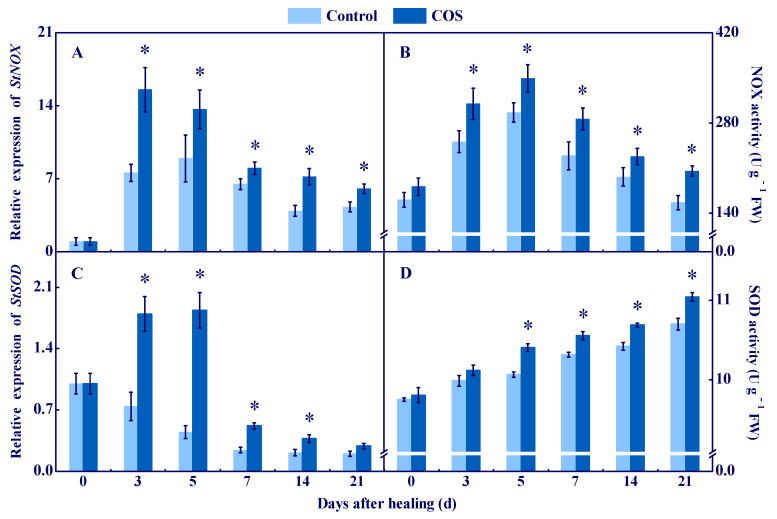
Effect of COS treatment on the relative expression of StNOX (**A**) and StSOD (**C**), and the activities of NOX (**B**) and SOD (**D**) at wounds of potato tubers during healing. Bars indicate standard error. Asterisks denote significant differences (*p* < 0.05).

**Figure 2 antioxidants-11-01791-f002:**
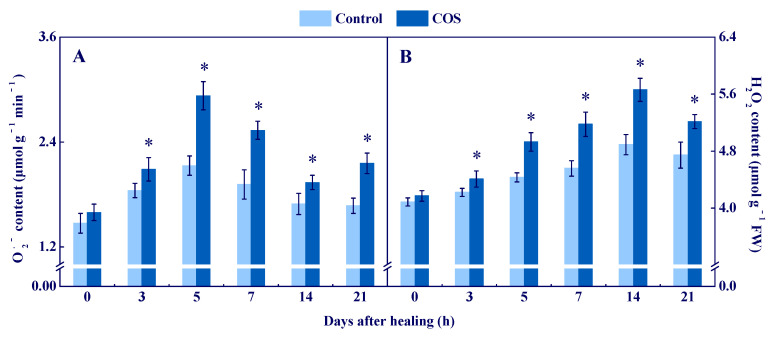
Effect of COS treatment on the contents of O_2_^●−^ (**A**) and H_2_O_2_ (**B**) at wounds of potato tubers during healing. Bars indicate standard error. Asterisks denote significant differences (*p* < 0.05).

**Figure 3 antioxidants-11-01791-f003:**
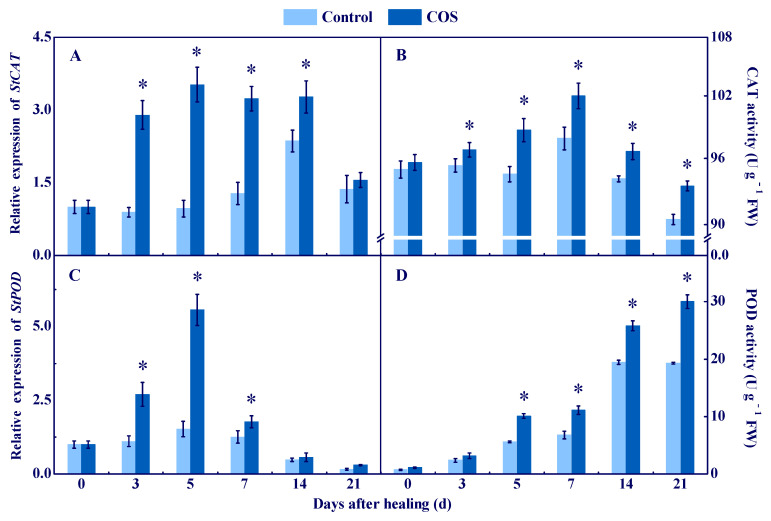
Effect of COS treatment on the relative expression of StCAT (**A**) and StPOD (**C**), and the activities of CAT (**B**) and POD (**D**) at wounds of potato tubers during healing. Bars indicate standard error. Asterisks denote significant differences (*p* < 0.05).

**Figure 4 antioxidants-11-01791-f004:**
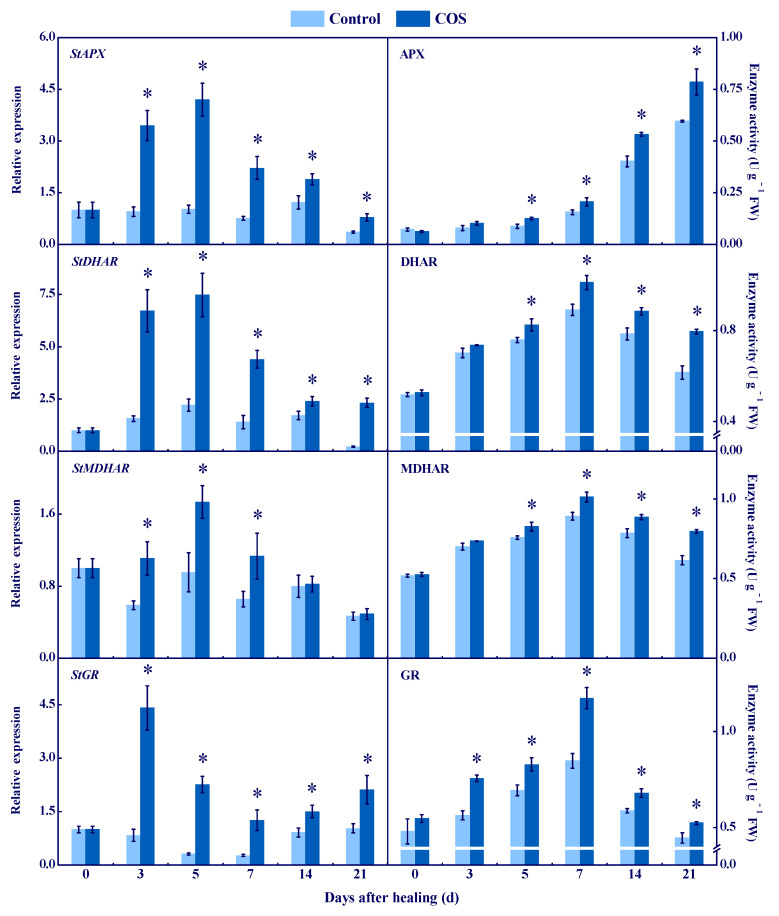
Effect of COS treatment on the relative expression of *StAPX*, *StDHAR*, *StMDHAR,* and *StGR*, and the activities of APX, DHAR, MDHAR, and GR at wounds of potato tubers during healing. Bars indicate standard error. Asterisks denote significant differences (*p* < 0.05).

**Figure 5 antioxidants-11-01791-f005:**
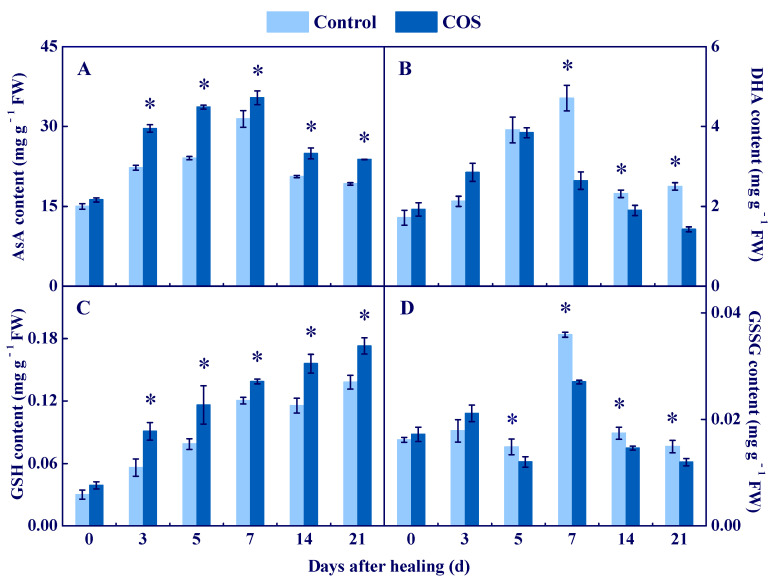
Effect of COS treatment on the contents of AsA (**A**), DHA (**B**), GSH (**C**), and GSSG (**D**) at wounds of potato tubers during healing. Bars indicate standard error. Asterisks denote significant differences (*p* < 0.05).

**Figure 6 antioxidants-11-01791-f006:**
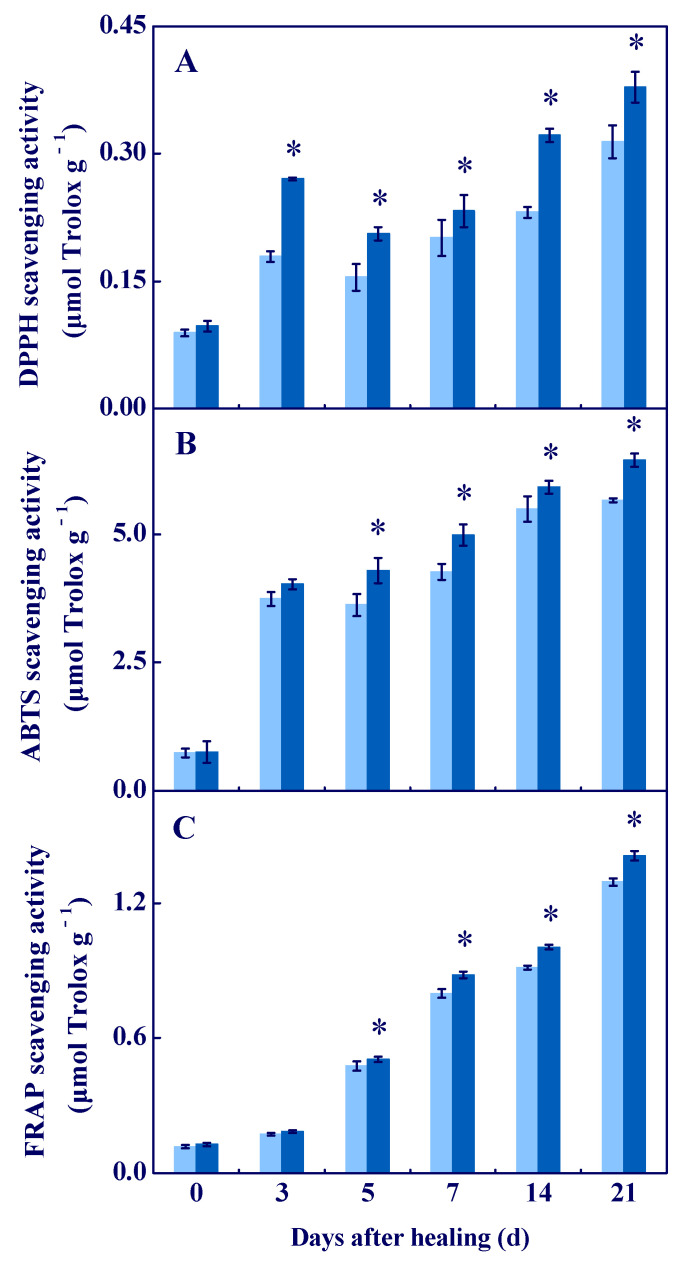
Effect of COS treatment on the scavenging ability of DPPH (**A**), ABTS^+^ (**B**), and FRAP (**C**) at wounds of potato tubers during healing. Bars indicate standard error. Asterisks denote significant differences (*p* < 0.05).

**Figure 7 antioxidants-11-01791-f007:**
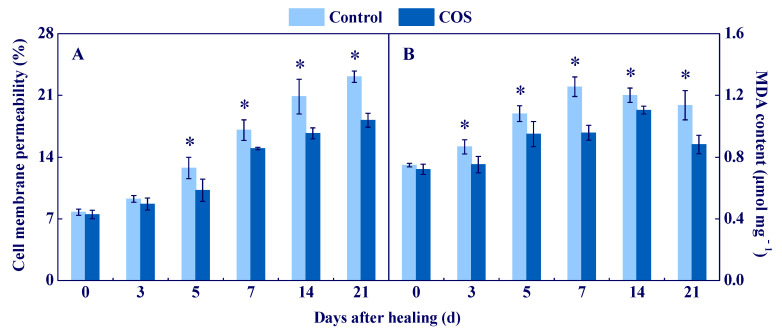
Effect of COS treatment on cell membrane permeability (**A**) and MDA content (**B**) at wounds of potato tubers during healing. Bars indicate standard error. Asterisks denote significant differences (*p* < 0.05).

**Table 1 antioxidants-11-01791-t001:** List of primer sequences used for Quantitative Real-Time PCR analysis.

Gene	Gene Accession Number	Primer Sequences (52032–3′)
*StNOX*	NM_001288375.1	Forward Primer: GTT TAC CTG GGC ATG AAC GC
		Reverse Primer: CTC CAC CAA TAC CGA CTC C
*StSOD*	AF354748	Forward Primer: GTT TGT GGC ACC ATC CTC TT
		Reverse Primer: GTG GTC CTG TTG ACA TGC AG
*StPOD*	M21334.1	Forward Primer: CAG CAA CCA AGG TAT AAT GTT T
		Reverse Primer: CGC GGA TGG AGG CAA GTC T
*StRCAT*	AY442179	Forward Primer: TGC CCT TCT ATT GTG GTT CC
		Reverse Primer: GAT GAG CAC ACT TTG GAG GA
*StAPX*	AB041343.1	Forward Primer: ACC AAT TGG CTG GTG TTG TT
		Reverse Primer: TCA CAA ACA CGT CCC TCA AA
*StDHAR*	DQ512964	Forward Primer: AGG TGA ACC CAG AAG GGA AA
		Pr Reverse imer: TAT TTT CGA GCC CAC AGA GG
*StMDHAR*	NM_001247084.2	Forward Primer: GCT GAT CCC AAC TCT GCA ACT
		Reverse Primer: CAC TCT CGA GGA ATG CAC CAA
*StGR*	X76533	Forward Primer: GGA TCC TCA TAC GGT GGA TG
		Reverse Primer: TTA GGC TTC GTT GGC AAA TC
*Efla*	AB061263.1	Forward Primer: CAA GGA TGA CCC AGC CAA G
		Reverse Primer: TTC CTT ACC TGA ACG CCT GT

## Data Availability

Data is contained within the article.

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
