# Peer review of "Chitooligosaccharide Maintained Cell Membrane Integrity by Regulating Reactive Oxygen Species Homeostasis at Wounds of Potato Tubers during Healing"

_antioxidants, 2022, doi:10.3390/antiox11091791_

Round 1

Reviewer 1 Report

The manuscript reported the effect of COS treatment increased resistance of reactive oxygen species (ROS) metabolism and antioxidant capacity in potato tubers during healing. It could be get the results that COS maintain cell membrane integrity by improving the antioxidant capacity in vitro, which contributes to maintaining cell membrane integrality at tuber wounds during healing. The manuscript is well organized but the data or writing needed to be improved. The English of this manuscript could be improved. Some other comment as following:

1.      It is not suitable to use ‘fruit’ in this manuscript. Because the material is potato tubes not fruit.

2.      The unit of GSH and GSSG contents should be use mg/g.

3.      How about the other quality of potato tubes after treatment, such as decay, color, firmness ?
